# Intraoperative Optical Coherence Tomography in the Management of Macular Holes: State of the Art and Future Perspectives

**DOI:** 10.3390/biomedicines10112873

**Published:** 2022-11-09

**Authors:** Filippo Confalonieri, Hanna Haave, Ragnheidur Bragadottir, Ingar Stene-Johansen, Xhevat Lumi, Lyubomyr Lytvynchuk, Goran Petrovski

**Affiliations:** 1Department of Ophthalmology, Oslo University Hospital, Kirkeveien 166, 0450 Oslo, Norway; 2Center for Eye Research and Innovative Diagnostics, Department of Ophthalmology, Institute for Clinical Medicine, University of Oslo, Kirkeveien 166, 0450 Oslo, Norway; 3Department of Biomedical Sciences, Humanitas University, Via Rita Levi Montalcini 4, Pieve Emanuele, 20090 Milan, Italy; 4Eye Hospital, University Medical Centre Ljubljana, Zaloška cesta 2, 1000 Ljubljana, Slovenia; 5Department of Ophthalmology, Justus Liebig University Giessen, University Hospital Giessen and Marburg GmbH, 35392 Giessen, Germany; 6Karl Landsteiner Institute for Retinal Research and Imaging, 1030 Vienna, Austria; 7Department of Ophthalmology, University of Split School of Medicine and University Hospital Centre, 21000 Split, Croatia

**Keywords:** intraoperative OCT, macular hole, vitreoretinal surgery, surgical technique

## Abstract

Background: Non-invasive diagnostic technologies in ophthalmology have substantially transformed contemporary clinical practice. Intraoperative optical coherence tomography (iOCT) systems have recently been used for various surgical interventions, including the treatment of full-thickness macular holes (FTMHs). Materials and Methods: We conducted a systematic review on the use of iOCT and its possible benefits in the management of FTMHs, following the Preferred Reporting Items for Systematic Reviews and Meta-Analyses guidelines (PRISMA). The level of evidence according to the Oxford Centre for Evidence-Based Medicine (OCEM) 2011 guidelines, and the quality of evidence according to the Grading of Recommendations Assessment, Development and Evaluation (GRADE) system, were assessed for all included articles. Results: 1131 articles were initially extracted, out of which 694 articles were obtained after duplicates were removed and their abstracts screened. A total of 65 articles was included for full-text review. Finally, 17 articles remained that fulfilled the inclusion criteria. Conclusions: Even though there is just a small number of studies with solid results, the use of iOCT in FTMH surgery may be a helpful tool for both novice and experienced surgeons planning and managing difficult cases. Additionally, it could be used for teaching reasons and for exploring novel surgical techniques.

## 1. Introduction

Pars plana vitrectomy (PPV) is the most performed vitreoretinal procedure, and there has been an increase in the number of patients undergoing PPV since the early 2000s in the Western world [1,2]. This type of surgery is used to treat a wide range of retinal disorders, including diseases of the vitreomacular interface, and has undergone substantial upgrading in recent years [3]. The emergence of minimally invasive procedures and the availability of cutting-edge technologies has made this treatment more successful and safer [4].

Intraoperative optical coherence tomography (iOCT) devices have recently been fully embedded into ophthalmic surgical microscopes, opening new perspectives and widening the possibilities of tissue imaging during vitreoretinal surgery, as well as during anterior segment surgery [5].

iOCT is a non-contact, real-time, high-resolution imaging technique that can assess the smallest and thinnest ocular structures intrasurgically, from the cornea up to the retina and the optic nerve, without an interruption of the surgical procedure (Figure 1 and Figure 2) [6]. However, the value of the real advantages of incorporating iOCT into routine clinical practice is still being debated [7,8].

A full-thickness macular hole (FTMH) is a common vitreomacular interface disorder that can cause significant central vision loss and reduce patients’ quality of life [9]. FTMHs are typically idiopathic; however, they can be associated with trauma, excessive myopia, vitreoretinal traction syndrome, proliferative diabetic retinopathy, age-related macular degeneration, or solar retinopathy [10]. Application of iOCT-assisted PPV for treatment of macular disorders, especially FTMH and its efficacy, has been reported [11]. The studies describe the methodology, handling, use, and advantages of three different iOCT systems: handheld, microscope-mounted, and microscope-integrated iOCT [11].

The purpose of this systematic review is to analyze and summarize current uses of iOCT in the management of FTMHs while evaluating the level and quality of the research included.

## 2. Materials and Methods

A systematic review was conducted and reported according to the Preferred Reporting Items for Systematic Reviews and Meta-Analyses (PRISMA) guidelines [12]. The review protocol was not recorded in the study design, but a registration number will be available for consultation. The methodology used consisted of a systematic search of all available articles exploring the use of iOCT in patients undergoing surgery for FTMH. To identify all relevant published articles, we performed a systematic literature search on 18 August 2022 using controlled vocabulary and test words for “vitreoretinal surgery” “vitrectomy” “retina tear” “macular hole’, “optical coherence tomography” and “intraoperative optical coherence tomography” in the electronic databases Ovid Medline, Embase (Ovid), Cochrane Register of Controlled Trials, and Cochrane Database of Systematic Reviews. The search was not restricted by language, publication type, study design, or date of publication. The complete search strategy is given in Appendix A.

Subsequently, the reference lists of all identified articles were examined manually to identify any potential study not selected by the electronic searches. After the preparation of the list of all electronic data, two reviewers (FC and HH) examined the titles and abstracts independently and identified relevant articles. Exclusion criteria were review studies, pilot studies, case series with fewer than 8 patients, case reports, photo essays, and studies written in languages other than English. Also excluded were studies performed on animal eyes, cadaveric eyes, and pediatric patients.

The same reviewers registered and selected the studies according to the inclusion and exclusion criteria by examining the full text of the articles. Any disagreement was assessed by consensus, and a third reviewer (GP) was consulted when necessary. No further unpublished data were obtained from the corresponding authors of all selected articles, which were analyzed to assess the level of evidence according to the Oxford Centre for Evidence-Based Medicine (OCEM) 2011 guidelines [13] and the quality of evidence according to the Grading of Recommendations Assessment, Development and Evaluation (GRADE) system [14].

## 3. Results

Figure 3 summarizes the flow diagram of the search approach, and the results of the analysis are collected and displayed in Table 1. A total of 1131 articles was initially extracted. Consequently, 694 articles were obtained after duplicates were removed and their abstracts were screened, and 65 articles could be included for the full-text review and more in-depth evaluation of inclusion/exclusion criteria. Finally, 17 articles remained that fulfilled all the inclusion criteria. The determining reasons behind each choice are summarized in Appendix B.

Table 1 summarizes the features, key outcomes, degree, and grade of existing evidence concerning the impact of iOCT in FTMH surgical management.

Due to the heterogeneity of the available data and the study designs, no data synthesis could be attained. As a result, the current review presents a qualitative analysis that is systematically conducted below.

Ray et al. conducted a retrospective analysis of all patients receiving FTMH surgery or epiretinal membrane (ERM) removal with non-microscope-integrated Spectral Domain iOCT (iSD-OCT) [31]. They utilized the Bioptigen system (Bioptigen, Inc; InVivoVue Clinic v1.2), which consists of an SD-OCT handheld device that requires a steady operator. Steadiness is required to obtain reproducible images of the macular region, which is a long and complicated task to complete with such an instrument. A total of 13 eyes of 13 patients who received PPV for FTMH and had iSD-OCT scan were recruited. Two patients displayed low-quality images and were excluded, whereas the remaining 11 eyes were subjected to quantitative analysis, which demonstrated the stability of the FTMH height and the hole diameter upon internal limiting membrane (ILM) peeling, as well as an increase in the diameter of the subretinal fluid under the macula in 10 out of 11 eyes (e.g., on average an 87% wider diameter). They concluded that the use of iSD-OCT provided new insights into the changes of the FTMH shape during surgery and that the device could be a useful tool in retinal surgery.

Ehlers et al., using iOCT, retrospectively assessed the intraoperative retinal architecture and FTMH geometric modifications that occur following surgical repair [30]. They used the microscope-mounted Bioptigen SDOIS system (Bioptigen Inc, Research Triangle Park, NC, USA). iOCT could successfully scan 21 eyes, out of which 19 had images with enough signal potency to enable quantitative analysis. Following ILM peeling, significant changes in FTMH shape could be observed, including increased FTMH volume, increased base area, and decreased top (inner or apex) area. Furthermore, enhanced subretinal hyporeflectance was observed by increasing the height between the inner/outer segment and RPE bands. The extent of the architectural modifications did not correspond with the peeling procedures or the surgeon’s experience. The visual results and FTMH closures were linked to the macular-hole algorithm readings and changes. They concluded that significant modifications occurred in the FTMH architecture and outer retinal configuration after ILM peeling. These changes could be observed only with iOCT imaging. Finally, the study demonstrated significant adjustments in the FTMH structure following surgical intervention, including increased FTMH volume, enlargement in the base (outer) size (area and diameter), and decrease in the internal area.

In a different study, Ehlers et al. performed an iOCT with the Bioptigen SDOIS system (Bioptigen, Inc., Research Triangle Park, NC, USA), which was connected to the surgical microscope through a customized microscope-mounting device [29]. This was a prospective, single-center, multi-surgeon investigation using iOCT. It included a case series of patients undergoing surgery for FTMH. The study was named “Prospective Assessment of Intraoperative and Perioperative OCT for Ophthalmic Surgery (PIONEER) study”. Imaging was examined at two time points during surgery: just before the start of the PPV (pre-incision scan) and after ILM peeling (post-peel scan). A total of 55 eyes that received effective FTMH operation was reported in the PIONEER trial. Thirteen eyes were rejected because of concomitant macular illness, one for a traumatic FTMH, one for absence of ILM peeling after surgery, and three for low quality in the acquired scans. As a result, 37 eyes from 36 patients were included. It was concluded that persistent subfoveal fluid following FTMH surgery is a frequent sign that seems to delay functional recovery but does not hinder the final outcome. The frequency of the presence of subfoveal fluid is very likely related to the changes in the outer layers of the retina caused by intraoperative manipulation during ILM peeling. The modifications included the ellipsoid zone and retinal pigmented epithelium (EZ-RPE) vertical height and the horizontal width of the subretinal hyporeflectivity. These findings were crucial in formulating new theories explaining FTMH closure through ILM peeling, such as altering photoreceptor/RPE adherence, which, by enhancing retinal mobility, can make the hole close completely.

Falkner-Radler et al. aimed at evaluating microscope-integrated iSD-OCT during macular surgery in a monocentric, prospective investigation [28]. It was performed using a high-definition OCT system, Cirrus HD (Carl Zeiss Meditec, Oberkochen, Jena, Germany), adapted to the optical pathway of the OPMI VISU 200 surgical microscope (Carl Zeiss Meditec, Oberkochen, Germany). The study compared the efficacy of iOCT imaging to the use of retinal dyes for ILM identification in macular surgery. The research comprised 51 individuals with ERM, 8 of whom had an associated lamellar macular hole, 11 of whom had vitreomacular traction, and 8 of whom had FTMH. One of the FTMH patients displayed a curled margin of the ILM in the iSD-OCT images, which made the iSD-OCT-assisted ILM peeling possible without requiring dye-assisted ILM staining. A macular ILM peeling with dye assistance was necessary for the remaining seven individuals. In all eight cases, preoperative iSD-OCT revealed a disrupted EZ and/or external limiting membrane (ELM) surrounding the FTMH. iSD-OCT scans after ILM peeling revealed an increased width of the FTMH base due to enhanced subretinal hyporeflectivity under the neighboring retina produced by the increased height between the EZ and the RPE. They concluded that iOCT was not inferior to retinal dyes in confirming successful removal of membranes. Nevertheless, the visualization of the flat membranes, as ILM often appears around the FTMH, was better after staining than with iOCT.

Riazi-Esfahani et al. studied the utility of a hand-held iSD-OCT in finding retinal structural alterations in 32 eyes during various steps of macular interventions for FTMH, ERM, and vitreomacular traction syndrome [27]. iOCT images were acquired using the iVue handheld SD-OCT (Optovue Inc., Fremont, CA, USA). The images were taken in three steps throughout the surgical procedure: before the induction of posterior vitreous detachment (PVD), after PVD induction, and after ILM peeling. Sixteen eyes of 16 patients underwent PPV for FTMH. One patient was diagnosed with a traumatic macular hole, whereas the other 15 had idiopathic FTMHs. In the 16 cases of FTMH, a decrease in FTMH apex (inner or top) diameter was observed following ILM peeling, whereas there were no major changes in the FTMH base diameter and height. These results are in contrast to those reported by Ray et al., who reported an increase in FTMH base diameter and no change in FTMH minimum diameter [31]. According to Riazi-Esfahani et al., the absence of ERM in most cases could have required less manipulation during surgery than in the study of Ray et al., which could have reduced the possibility of subretinal fluid accumulation and retinal detachment. They concluded that the success rate of closure of FTMHs may benefit from additional intraoperative measurements and could be anticipated by a decrease in FTMH diameters demonstrated by iOCT.

Uchida et al. aimed at investigating acute retinal alterations identified with iOCT immediately after surgical operation with the Finesse Flex Loop for vitreoretinal interface disorders [26]. This study is part of the Determination of feasibility of Intraoperative Spectral domain microscope Combined/integrated OCT Visualization during *En face* Retinal and ophthalmic surgery (DISCOVER) study, a prospective study analyzing the feasibility and use of iOCT integrated into a surgical microscope [32]. This study used a prototype Rescan 700 (Carl ZeissMeditec, Oberkochen, Germany) iOCT device connected to a Lumera 700 (Carl Zeiss Meditec) microscope. Altogether, 34 eyes of 34 patients with a mean age of 72.7 years were studied, 25 of whom were women. The intraoperative diagnosis was FTMH in 21 eyes (62%) and ERM in 13 (38%) cases. In all the procedures, an indocyanine green-assisted ILM peel was started by using the membrane loop. They concluded that the membrane loop aided in the commencement of the ILM peeling and provided clinically adequate safety throughout the macular surgery with no major intraoperative adverse events. Following membrane-peeling treatments, the iOCT revealed a low rate of retinal changes.

Bruyère et al. reported on the experience provided by iOCT imaging during vitreomacular surgery in highly myopic eyes [25]. They performed a retrospective consecutive case-series analysis on highly myopic eyes that received macular surgery for ERM, FTMH, and myopic foveoschisis using iOCT. Ten eyes were diagnosed with FTMH. The study conducted a qualitative and quantitative evaluation of retinal alterations following each phase of the surgical procedure, including the discovery of persisting epiretinal structures, new retinal tears, central macular thickness, and FTMH sizes. iSD-OCT images were acquired throughout operation with the Rescan™ 700 system (Carl Zeiss Meditec, Oberkochen, Germany). The measurements of minimal horizontal diameter of FTMHs obtained before and after ILM peeling were not significantly different. Similarly, the FTMH base diameters obtained before and after ILM peeling were not significantly different. The study also demonstrated that the iOCT acquisitions passing through the optic disc and the macular region might represent a useful screening step in the detection of PVD in highly myopic eyes before using triamcinolone acetonide. Furthermore, the study concluded that iOCT enhances surgeons’ ability to detect ERM and ILM as well as the presence of retinal holes.

Kumar et al. described a new intraoperative sign during PPV for FTMH using an iSD-OCT in an attempt to anticipate the likelihood of closure of FTMHs [24]. This retrospective interventional study included 25 eyes of 25 patients with FTMH who received a 25-gauge PPV over a 16-month period at a tertiary referral hospital by the same surgeon. Before and after ILM peeling, all eyes were evaluated using the iSD-OCT Rescan™ 700 system (Carl Zeiss Meditec, Oberkochen, Germany). The study concluded that the hole-door sign, defined as the visualization of remaining tissue on the margin of the FTMH following ILM peeling, was a reliable predictor of postoperative hole closure. All eyes with the hole-door sign reached successful closure. However, the precise structure of such tissue pieces remains unknown. They are probably not vitreous remnants because no tissue fragment was observed following PVD induction and before ILM peeling began. They hypothesized that the hole-door sign might have been generated by superfluous retinal tissue, subclinical ERMs, or minute remnants of ILM connected to the hole’s borders.

Runkle et al. aimed at assessing the correlation between the dissociated optic-nerve-fiber layer (DONFL) and the intraoperative membrane-peeling retinal modifications as visualized by iOCT, as well as the DONFL’s functional repercussion [23]. This study examined patients from the PIONEER study. Intraoperative images were acquired with the microscope-integrated Bioptigen Envisu iOCT system (Bioptigen, Research Triangle Park, NC, USA). Overall, 95 eyes were included, and the preoperative diagnoses included ERM in 54 eyes (57%) and FTMH in 41 eyes (43%). They concluded that the development of DONFL was related to the surgical indication for FTMH. On the contrary, the development of DONFL was not related to the choice of surgical instrument (forceps alone or a combination of forceps with other instruments). However, it was uncertain whether the FTMH condition per se or the surgical procedure, such as ILM peeling, were the most important element influencing DONFL appearance. Overall, their data implied that intraoperative damage to the inner retina, possibly during ILM peeling, might have a crucial role in the DONFL appearance.

Ehlers et al. aimed at establishing a predictive model of FTMH closure rate and velocity using iOCT [22] in a post hoc study on eyes that received surgery for FTMH. The Bioptigen SDOIS system (Bioptigen, Inc., Research Triangle Park, NC, USA) was used for iOCT imaging. A total of 62 eyes was identified that received surgery for FTMH. Ten eyes could not be included because of poor OCT image resolution through the gas in the vitreous chamber, and 15 eyes had to be excluded due to insufficient iOCT data collection. Finally, 37 eyes from 37 patients were included in the final analysis; 32 (86%) out of 37 eyes showed FTMH closure at postoperative day one. At the 3-month control, FTMH closure was reached in 35 sets of eyes (95%). Following a multivariate logistic-regression analysis, seven covariates could be formulated, and they were proposed as predictive variables for FTMH closure rate and velocity: age, EZ-RPE expansion following ILM peeling, preoperative minimal diameter, post-ILM peeling, FTMH height and change in FTMH volume, minimum FTMH width, and FTMH depth. They concluded that iOCT has the potential to become a crucial tool for predicting FTMH closure rate and velocity.

Inoue et al. aimed at determining the importance of the correlation between iOCT findings and the postoperative retinal anatomical and functional characteristics in eyes with FTMH [21]. Thirty-three eyes with FTMH were included and intraoperatively analyzed with iOCT system Rescan™ 700 (Carl Zeiss Meditec, Oberkochen, Germany) to find residual fragments of ILM/ERM at the margins of the hole. Two groups were created based on the iOCT features: one group with residual fragments (the residual group) and one group without residual fragments (the non-residual group). Residual fragments were found in 22 eyes (67%), including three with residual ILM fragments, whereas such fragments were absent in 11 eyes. Age, gender, pre-operative visual acuity, axial length, and refractive errors did not differ substantially across the groups. In the preoperative OCT images, the presence of remaining fragments was substantially linked to the existence of ERM. Postoperatively, all FTMHs were closed. At 3 and 6 months, the residual group’s postoperative visual acuity was considerably impaired. They concluded that the residual fragments found at the margin of the FTMH by iOCT might be the hyperreflective material seen in closed FTMHs and are predictors of scarce postoperative visual outcome.

Lytvynchuk et al. assessed the efficacy of iOCT imaging for the inverted ILM flap technique (IILMFT) in large FTMHs [20]. It was a non-randomized, prospective, observational study conducted on eight eyes of seven patients diagnosed with large (5 patients, 7 eyes) and recurrent (1 patient, 1 eye) FTMHs. The microscope-incorporated iSD-OCT system Rescan™ 700 (Carl Zeiss Meditech, Oberkochen, Germany) and EnFocus™ UltraHD (Leica Mikrosysteme Vertrieb GmbH, Wetzlar, Germany) were used. Despite the shadowing caused by the steel instruments, the distance between the retinal layers and the instrument tips could be monitored and controlled. The study concluded that the iatrogenic influence on the retina can be detected by iOCT in surgical phases such as mechanical apposition and ILM peeling of FTMH margins (depression and appearance of hyporeflective zones). Towards the end of the procedure, after fluid-air exchange, iSD-OCT imaging could validate the appropriate location of the inverted ILM flap.

Lorusso et al. aimed at assessing the visual outcome and closure rate of patients with FTMH retrospectively, following confirmation of the FTMH closure by iOCT and after a short-term (12–24 h) postoperative face-down posturing (FDP). The secondary objective was to determine the correlation between iOCT and postoperative OCT at day 1 after surgery [19]. Twenty-nine eyes of 29 patients with FTMH were enrolled in the study. FTMH closure was confirmed in all patients intraoperatively with the RESCAN 700 system (Zeiss, Oberkochen, Germany). The mean number of hours of FDP was 18 ± 2.6 h, and at 3 months the FTMH closure rate was 93%; two eyes received secondary FTMH repair surgery. The final FTMH closure rate was 100%. It was concluded that iOCT-based confirmation of the FTMH closure, having a high closure rate and no additional complications, may be a reliable and effective method for advising short-term FDP following the operation.

Tao et al. aimed at retrospectively investigating the ability of iOCT to assess different FTMH margin structures and anticipate the recovery of visual function and retinal architecture [18]. Fifty-three eyes were included and intraoperatively scanned with Optovue iVue OCT System (Optovue, Inc., Fremont, CA, USA). The FTMHs were categorized into three groups based on the appearance and shape of the hole margins. The first group (Hole-Door group) displayed vertical columns of tissue that extended into the vitreous chamber after ILM peeling. The second group (Foveal Flap group) preoperatively displayed a foveal flap that was adherent to the hole margins following ILM peeling. The third group (Negative group) displayed neither a hole-door nor a foveal flap. At 6 months following the surgical procedure, the retinal structural recovery and visual function were described and a comparison was made among the three groups. All eyes showed FTMH anatomical closure, and the postoperative best corrected visual acuity (BCVA) showed a significant improvement compared to the preoperative BCVA. The Hole-Door and the Foveal Flap groups showed a significantly better final BCVA and recovery of the ELM compared to the Negative group. Analyzing the subpopulation in which the FTMH diameter was ≤ 400 μm, no significant differences in ELM restoration, EZ, or BCVA was demonstrated in any of the three groups. In the subpopulation with FTMH > 400 μm, the Hole-Door and Foveal Flap groups showed a significantly better final visual acuity and restoration of ELM compared to the Negative group. They concluded that iOCT during FTMH surgery can confirm the presence of the hole margins structured as hole-door, foveal flaps, or neither, and that the images obtained by iOCT can supply the clinician with important predictive information for postoperative retinal structural and visual function improvement of large FTMHs.

Nishitsuka et al. aimed at observing intraoperative changes in FTMH morphology using iOCT [17]. They included 10 eyes from 10 patients undergoing surgical treatment for FTMH detected intraoperatively by the Rescan™ 700 system (Carl Zeiss Meditech, Oberkochen, Germany). The mean shortest FTMH diameter was significantly reduced following fluid–gas exchange, but no patients showed iOCT-confirmed closure of the FTMH. Nevertheless, FTMH closure was achieved in each patient. They concluded that a reduction of the FTMH diameter after fluid–air exchange is a crucial modification that can be demonstrated by iOCT.

Tao et al. aimed at reporting a four-year assessment of feasibility and utility of iOCT for various vitreoretinal diseases in China [16]. The Optovue iVue OCT System (Optovue, Inc., Fremont, CA, USA) was used. A total of 339 eyes successfully received an iOCT scan, of which 59 were FTMHs. Of these, 51 patients were treated by conventional ILM peeling, five patients underwent an inverted-flap technique, and three patients had a free-flap technique. In the 51 patients that received a conventional ILM peeling, the areas of the hole were all reduced intraoperatively. Twenty cases presented the hole-door phenomenon. The inverted ILM flap could be identified at the iOCT during the procedure in all cases for both the ILM inverted-flap and the free-flap techniques. They concluded that the iOCT plays an important role during retinal surgery and can serve as a guide intraoperatively during FTMH repair.

Yee et al. aimed at characterizing the clinical outcomes and the role of iOCT-assisted FTMH repair [15]. This research included every single patient enrolled in the DISCOVER study who received a surgical procedure for FTMH. The images were acquired through either the Rescan™ 700 (Carl Zeiss Meditech, Oberkochen, Germany) or the EnFocus (Leica/Bioptigen, Research Triangle Park, NC, USA) system. Eighty-four eyes were included in this study. In 43 patients (51%), the operators affirmed that iOCT provided useful information (e.g., assuring the release of tractions on the macular region and finding unrecognized residual membranes). In 10 patients (12%), the information provided by the iOCT modified the surgical plan and decision-making process. Postoperative day-one trans-tamponade OCT confirmed the margin apposition and the hole closure in 74% of eyes (21/26). Five holes were still open on postoperative day one, but they closed following positioning. The single-surgery FTMH anatomical success rate was accomplished in 97.6% of the operated eyes. Only one persistent FTMH required a subsequent surgical repair to reach the anatomical success, and it determined an overall final anatomical success rate of 98.8%. One single FTMH remained open, but, since it was both chronic and large, it was decided that no surgery was required. They concluded that iOCT can be an important new tool in FTMH surgery, affecting the surgical decision-making process. Moreover, iOCT-assisted FTMH surgery showed significant improvement in BCVA and a high success rate of closure.

## 4. Discussion

A step forward towards safer and more effective surgery may be represented by the introduction of surgical microscopes with integrated iOCT. A variety of iOCT devices may already be connected to an ocular microscope, giving helpful information to both anterior-segment and vitreoretinal surgeons [5,6,33,34].

This technology, when applied to vitreoretinal surgical procedures, enables the direct visualization of vitreoretinal structural relations during the entire surgical procedure, allowing the assessment of surgical planes, guiding of surgical steps, and aiding in the detection of intraoperative complications, ultimately influencing surgical decision-making [35,36].

When dealing with FTMH, iOCT has been proposed as a tool capable of guiding a more precise and rigorous intraoperative approach since the first study by Ray et al. in 2011 [31]. The outcomes of the PIONEER study and the DISCOVER study, two prospective case series, could generate a large part of the available literature on iOCT applied to FTMH surgery [15,22,23,26,33].

Both Ray et al. [31] and Ehlers et al. [30] found significant alterations in FTMH geometry after surgical manipulation, including increased total FTMH volume and, specifically, an increase in the base size (area and diameter in the outer retina), but a decrease in the top (apex or inner) area of the hole immediately after ILM peeling. These results allow us to expect that more significant intraoperative changes in FTMH morphology might reflect an increased hole laxity and indicate an increase in the probability of FTMH closure. Their results might be considered only partially coherent with the subsequent study by Riazi-Esfahani et al. [27] and Nishitsuka et al. [17], who also found a decrease in the inner area of the FTMHs post-ILM peeling, whereas they found no increase in the base diameter. Furthermore, Nishitsuka et al. described that the mean minimum diameter of FTMH decreased significantly after fluid–gas exchange, highlighting the importance of gas-surface tension, even though the patients did not demonstrate intraoperative FTMH closure. Nevertheless, all holes were closed on postoperative day one, which is in agreement with what has been recently postulated by Wu et al. [37]. These findings suggest that FTMH closure occurs after surgery, rather than during surgery, likely as a result of postoperative Müller-cell migration.

However, it appears possible to achieve intraoperative closure of FTMHs after ILM peeling, as demonstrated by Lorusso et al. [19], and this could be considered a positive prognostic factor and the end point of the previously elucidated reduction in the inner FTMH’s area after ILM peeling. According to Lorusso et al., confirming the intraoperative FTMH closure could be regarded as a sign to prescribe only a 12–24 h postoperative posturing to the patient. These findings gain even more importance if correlated with the current trend of evidence showing that strict FDP is not necessary for achieving FTMH closure, even for large holes [38,39].

Kumar et al. [24] and Inoue et al. [21] reported that iOCT can detect remnants of ILM at the edge of the FTMH after vitrectomy and ILM peeling. Kumar et al. named these fragments a “hole-door sign” and although the presence of such a sign was significantly correlated with the closure rate of FTMHs in the first study, the second study showed that, despite the possibility of achieving better anatomical success, the interposed tissue may lead to reduced BCVA postoperatively. The remaining tissue could be identified by foveal hyperreflectivity on postoperative OCT. Pathophysiologically, the remaining fragments in the closed FTMH may turn into hyperreflective inner retinal tissue and reshape postoperatively to compensate for the closed FTMH’s inner foveal defect. Despite the good anatomical healing process of inner retinal layers in closed FTMHs, the postoperative repair of the EZ defect and functional visual recovery in eyes with residual pieces can be impaired. This might be explained by the contractile capability of the leftover pieces, which could hinder the healing process of the outer retina. These conclusions are only partially in accordance with what was subsequently described by Tao et al. [18], leaving the matter unsettled and requiring further research to better understand the correlation between intraoperative tissue configuration and postoperative anatomical and functional outcomes. Further research in this field is also necessary to elucidate the correlation between FTMH healing after the inverted ILM flap technique and the functional outcome, especially considering recent studies suggesting that this technique does not have additional benefits for small–medium-size FTMHs, and may delay recovery of retinal integrity [40,41].

Other studies, such as the one by Falkner-Radler et al. [28], focused mainly on the comparison between iOCT and retinal dyes. They showed the non-inferiority of iSD-OCT in comparison to retinal dyes to visualize non-flat ERMs. Indeed, following staining, the visibility of flat membranes was better, highlighting that the iOCT technology currently available cannot completely substitute the role of vital dyes, especially in FTMH surgery, where ILM usually appears as a flat membrane.

Focusing on flex-loop intraoperative retina alteration, Uchida et al. [26] showed that the membrane loop instrument made it easy to start the ILM peeling, and it provided the surgeon with a good safety profile without any serious intraoperative complications during FTMH surgery. This technique of ILM-peeling initiation can be considered safe and effective because the iOCT demonstrated that the retina undergoes only minor structural modification during the scraping maneuver.

In the context of FTMH and high myopia, Bruyère et al. [25] showed that iOCT allowed for visualization of remnants of the posterior vitreous cortex. This is important, since myopic eyes often present with vitreoschisis, which can be difficult to visualize both preoperatively with OCT and intraoperatively without staining with triamcinolone [42]. Furthermore, iOCT could help assess undetected retinal holes and ERM peeling, making this technology particularly useful for guiding the surgeon in approaching myopic eyes with challenging anatomy.

In order to elucidate the mechanisms underlying the development of DONFL, Runkle et al. [23] showed that when the inner retina tended to thicken on iOCT, it was correlated to the appearance of DONFL. This finding suggests that an intraoperative traction or trauma in general to the inner retina can lead to damage to the retina that can appear morphologically as DONFL. As iOCT technology is refined, it could definitely acquire a potential role in preventing traumatic retina procedures from happening, and it may help develop atraumatic membrane-peeling techniques.

Ehlers et al. [22] described for the first time a predictive, iOCT-based model for FTMH closure. Consistent with Ip et al. [43], Ullrich et al. [44] and Wakely et al. [45], they found that preoperative minimal diameter of FTMH was a robust positive predictor for hole closure. However, they did not find any other predictor associated with preoperative OCT variables. Since the morphology of FTMHs and the ultrastructure of retinal layers are modified after ILM peeling, preoperative OCT variables (except for the minimal inner diameter) might have less impact on FTMH postoperative outcome than iOCT variables that can be acquired after ILM peeling. Endowing the vitreoretinal surgeon with iOCT technology can provide information on the retinal-tissue response to the ILM peeling, allowing for a more reliable prognosis. Furthermore, Ehlers et al. described six other statistically significant variables that were different in patients with successful and failed FTMH closure on postoperative day one: age, post-ILM peeling EZ-RPE expansion, post-ILM peeling depth, change in volume, change in minimum width, and change in depth. Because five out of the seven identified variables were only measurable with iOCT, we can affirm that this new technology can provide a unique benefit for the prediction of FTMH closure.

Lytvynchuk et al. [20] assessed the efficacy of dynamic iSD-OCT imaging for the inverted ILM flap technique in large FTMH surgery. This technology enables real-time imaging of the whole procedure, including viewing of the FTMH, vitreoretinal tools, and all processes of inverted ILM flap development, despite the shadowing caused by the steel instruments, one of the major limitations in retina visualization with iOCT. In the future, development of new instruments may allow for a better perception of the distance between the instrument tips and the retinal layers. Lytvynchuk et al. also showed that the iatrogenic influence on the retina was detected by dynamic imaging of surgical steps such as ILM peeling and mechanical apposition of FTMH edges (depression and appearance of hyporeflective zones). Such feedback could influence the decision-making process of the surgeon by allowing for immediate anatomical feedback on the technique utilized. Moreover, since at the very end of the procedure, following fluid–air exchange, intraoperative imaging can validate the appropriate location of the inverted ILM flap, iOCT might prove to be especially useful in challenging situations when the inverted ILM flap technique is not positioned properly or may fail. In fact, one main concern with the ILM flap technique is flap displacement, especially during fluid–air exchange [46]. A more widespread availability of iOCT in the future could contribute to making FTMH repair with the inverted ILM flap technique safer and more predictable. The efficacy of the iOCT approach during the inverted ILM flap technique was also reported by Maier et al., who noted it to be useful in intraoperative identification of retinal microstructure, making the procedure safer and more controlled [47].

Yee et al. [15] suggested that iOCT may have an important role in FTMH surgery, making surgical decisions more accurate. As iOCT-assisted FTMH surgery in their study resulted in significant improvement in BCVA and a high single-surgery success rate, future prospective studies could validate this finding to determine whether the single-surgery success rate of iOCT-assisted surgery is superior to conventional surgery.

Our systematic review has limitations: some of the reviewed literature was produced in languages other than English and therefore not included. Furthermore, the heterogeneity of the analyzed articles did not allow for direct comparison of the results and production of meta-analysis. This diversity stems from the iOCT devices used, the populations studied, the experience of the surgeons, and the techniques used during surgery. 

## 5. Future Perspectives and Technologies

Even though iOCT has made significant advancements in recent years, more work has to be done to surpass the present challenges to seamlessly integrate image-guided vitreoretinal surgery. Automated segmentation software, OCT-optimized surgical equipment, software interfaces, a thorough platform for virtual-reality visualization, and robot-assisted iOCT-guided surgeries are only a few of the areas in which there is a demand and potential for innovation [6,48].

When it comes to FTMH, the future development of this technology should aim for a 3D- or 4D- robot-assisted interface capable of seamlessly orienting the surgeon in the dissection of the ILM and epiretinal membranes. Due to the light-scattering and shadowing properties of metal, current metallic surgical instruments provide challenges for intraoperative OCT devices [32]. The development of new OCT-compatible instruments may allow the membranes that coaxially underlie the intraocular devices to be visualized and manipulated. Some evidence suggests the opportunity to implement intraocular OCT-endoprobe devices as an alternative [49]. In the future, the anatomy of FTMH should be approached through a segmentation of the retinal layers, automatically and volumetrically reconstructed thanks to an ad hoc software and presented to the surgeon as 3D or 4D virtual reality. New technologies will arise that incorporate swept-source high-resolution OCT technologies into classical and “heads-up” surgical microscopes. The dynamic assessment of the repercussions of the surgical maneuvers on retinal tissue might lead to new insights into the biomechanical response of the retina to ILM peeling and, therefore, to better guiding the surgeon. New advancements in iOCT technology are needed and could contribute to settling controversies such as the need for a customizable size of ILM peeling, the technique of choice for each case (e.g., inverted ILM flap technique and, for that matter, the subtype: temporal, cabbage leaf, or any other) and the control of factors associated with anatomic failure after FTMH surgery [50,51].

Despite the high potential, it is still necessary to validate the total value of iOCT for patient outcomes. Despite studies showing a definite influence on surgical decision-making, it is unclear how surgery will ultimately affect patient outcomes. To determine the total benefit more accurately, randomized masked prospective studies for disease-specific outcomes are necessary. Planning for these investigations is still ongoing.

## 6. Conclusions

Despite the fact that there are just a few studies with conclusive findings, the application of iOCT in FTMH surgery is a useful tool for both beginner and experienced surgeons planning and treating challenging cases, as well as for teaching purposes. More scientific evidence is required in order to validate the controversies regarding the interpretation of data deriving from the iOCT technology applied to FTMHs and to prove the real advantage in terms of diagnostics, prognosis, and clinical outcomes.

## Figures and Tables

**Figure 1 biomedicines-10-02873-f001:**
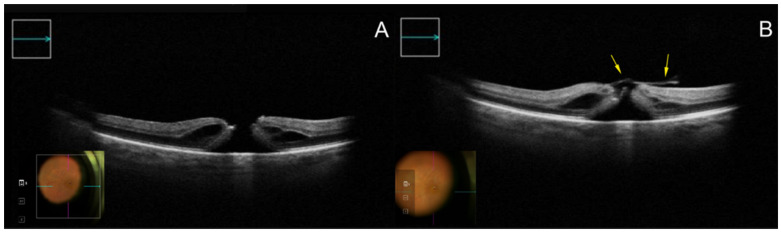
Appearance of an FTMH on iOCT before (**A**) and after (**B**) preparation of the temporal inverted ILM flap (yellow arrows) viewed through a horizontal cross-section (green arrow). Abbreviations: FTMH, full-thickness macular hole; iOCT, intraoperative optical coherence tomography; ILM, internal limiting membrane.

**Figure 2 biomedicines-10-02873-f002:**
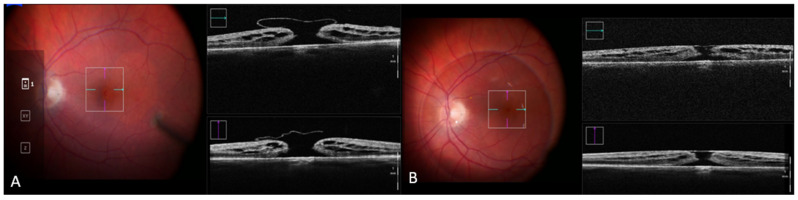
Appearance of an FTMH on iOCT after ILM peeling and inverted flap before (**A**) and after (**B**) PFCL injection viewed through a horizontal (green arrow) and vertical (purple arrow) cross-section. Notice that the ILM flap flattens after PFCL injection and that the diameter of the FTMH is reduced. Abbreviations: FTMH, full-thickness macular hole; iOCT, intraoperative optical coherence tomography; ILM, internal limiting membrane; PFCL, perfluorocarbon liquid.

**Figure 3 biomedicines-10-02873-f003:**
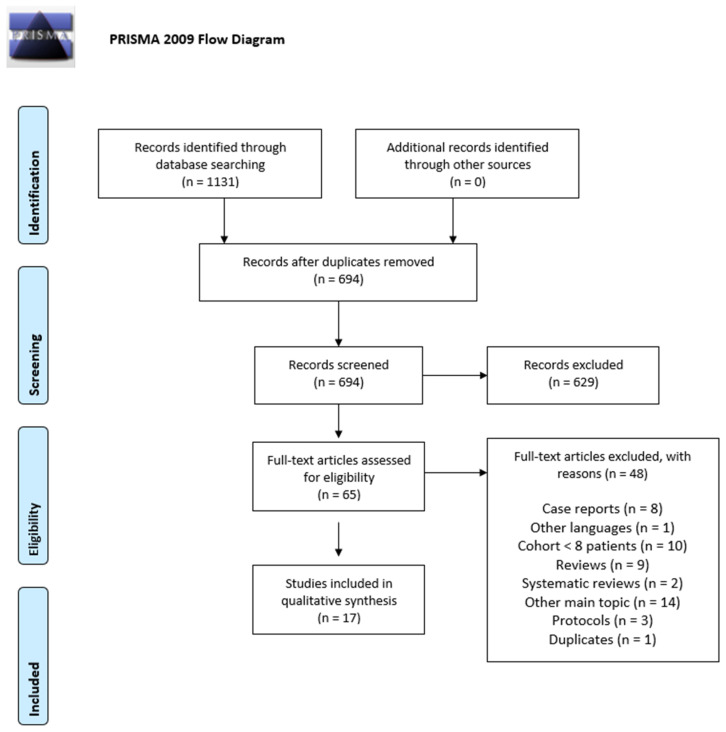
Flow diagram of the study according to Preferred Reporting Items for Systematic Reviews and Meta-Analyses (PRIMA) guidelines [12].

**Table 1 biomedicines-10-02873-t001:** Characteristics, quality, and level of evidence of the included studies and features of the iOCT cited by these articles.

Author (et al.)	Year	Study Design	Study Sample (Eyes)	Type of Surgery	iOCT Specifications	Grade ^1^	Level ^2^
**Yee [15]**	2021	Post hoc	84	PPV ILM peeling	Rescan 700/ EnFocus	Moderate	3
**Tao [16]**	2021	Retrospective	339	PPV ILM peeling	Adoptivue iOCT	Low	4
**Nisihitsuka [17]**	2021	Retrospective	10	PPV ILM peeling	Rescan 700	Very low	4
**Tao [18]**	2020	Retrospective	53	PPV ILM peeling	Adoptivue iOCT	Low	4
**Lorusso [19]**	2020	Retrospective	29	PPV ILM peeling	Rescan 700	Low	4
**Lytvynchuk [20]**	2019	Prospective, non-randomized, observational	8	PPV ILM peeling/ flap technique	Rescan 700/ EnFocus UltraHD	Very low	4
**Inoue [21]**	2019	Retrospective, case-control	33	PPV ILM peeling	Rescan 700	Low	4
**Ehlers [22]**	2019	Post hoc	37	PPV ILM peeling	Bioptigen SDOIS	Low	4
**Runkle [23]**	2018	Post hoc	95	PPV ILM peeling	Bioptigen Envisu	Moderate	4
**Kumar [24]**	2018	Retrospective, interventional	25	PPV ILM peeling	Rescan 700	Low	4
**Bruyere [25]**	2018	Retrospective, observational	22	PPV ILM peeling	Rescan 700, integrated HD OCT	Low	4
**Uchida [26]**	2017	Post hoc	34	PPV ILM peeling/flap technique	Rescan 700, Lumera 700	Low	4
**Razi-Esfahani [27]**	2015	Case series	32	PPV ILM peeling	iVue hand-held SD-OCT	Low	4
**Falkner-Radler [28]**	2015	Prospective, interventional	70	PPVILM peeling	Carl Zeiss Meditec/Cirrus	Moderate	4
**Ehlers [29]**	2014	Retrospective	21	PPV ILM peeling	Bioptigen SDOIS	Low	4
**Ehlers [30]**	2014	Prospective	37	PPV ILM peeling	Bioptigen SDOIS	Low	4
**Ray [31]**	2011	Retrospective	25 eyes (13 MH)	PPV ILM peeling	Bioptigen	Low	4

^1^ Quality of evidence according to the Grading of Recommendations Assessment, Development and Evaluation (GRADE) system [14]. ^2^ Level of evidence according to the Oxford Centre for Evidence-Based Medicine (OCEM) 2011 guidelines [13]. Abbreviations: PPV, pars plana vitrectomy; ILM, internal limiting membrane; MH, macular hole; iOCT, intraoperative optical coherence tomography.

## Data Availability

Data are available on reasonable request by the corresponding author.

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
