# Peer review of "Intraoperative Optical Coherence Tomography in the Management of Macular Holes: State of the Art and Future Perspectives"

_biomedicines, 2022, doi:10.3390/biomedicines10112873_

Round 1

Reviewer 1 Report

The paper entitled “Intraoperative Optical Coherence Tomography in the Management of Macular Holes: State of the Art and Future Perspectives” is an interesting descriptive systematic review of the use of iOCT in the management of full-thickness macular holes (FTMHs). The manuscript is interesting, innovative, and of potential clinical interest.

Patients with FTMHs tend to be handled based on the surgeon’s preferences and local retinal departmental guidelines. The advantage of iOCT is a non-contact, real-time, high-resolution imaging technique that can assess structures intrasurgically without an interruption of the surgical procedure. The systematic study shows that the use of iOCT in FTMH surgery may be a helpful tool for novice expert surgeons in managing difficult cases. This technology can also prove to be useful  for teaching reasons and assessing new surgical techniques. The 17 articles included in the systematic review have been summarized with appropriate take-home messages. This review can assist in the standardization of treatment based on signs, symptoms, and OCT imaging results.

The study has been correctly planned. It is well-written and of clinical interest. The study provides objective results, which add to the current literature in this field. Figures and tables are pertinent and provide good examples of macular pathologies. The references are numerous and are based on current imaging tools and surgical techniques. 

Author Response

Thank you so much, it is great to see your positive response.

Reviewer 2 Report

The authors presented a systematic review onthe use of Intraoperative optical coherence tomography and its possible benefits in the management of full thickness macular holes. The manuscript is with merit and the results are worth reporting but authors should address the following comments before publication:* Figures legends: the explanations for the abbreviations should be provided* In general, the authors should add a comma as thousand separator (i.e. line 107: 1131 articles should be 1,131 article), please revise the manuscript correspondingly* Table 1: it would be very useful if the authors added the number of the reference in the reference list for each study listed in the table for immediate reference for the readers* The authors should add a limitation section in which they present the limitation of their review

Author Response

Dear Reviewer,

  • The explanations for the abbreviations have now been provided.
  • A comma has been added as a thousand separator.
  • Table 1: the numbers of the references have been added for immediate reference to the readers.
  • A limitation section has been added.

Thank you and we hope our answers will be acceptable to the Reviewer in the form provided.